# Use of the Dialkylcarbamoylchloride Dressing in the Care of Central Venous Access Exit Site in a Pediatric and Neonatal Population

**DOI:** 10.3390/diagnostics13091520

**Published:** 2023-04-23

**Authors:** Giorgio Lamberti, Vincenzo Domenichelli, Simona Straziuso, Gabriella Pelusi, Miria Natile, Gina Ancora, Giovanni Barone

**Affiliations:** 1Pediatric Surgery, Infermi Hospital, AUSL della Romagna, 47923 Rimini, Italy; giorgio.lamberti@auslromagna.it (G.L.);; 2Neonatal Intensive Care Unit, Infermi Hospital, AUSL della Romagna, 47923 Rimini, Italygbarone85@yahoo.it (G.B.)

**Keywords:** central venous catheter, exit site, dialkylcarbamoylchloride, dressing, exit site infection

## Abstract

Dialkylcarbamoylchloride dressing is a fatty acid derivative that has been shown in vitro to bind a number of pathogenic microorganisms. The purpose of this prospective study was to evaluate the safety and the efficacy of this technology in the care of the exit site of central venous catheter in a paediatric and neonatal population. Methods: The study was conducted from September 2020 to December 2022 at the Infermi Hospital in Rimini. Central venous catheters were placed using the SIC bundle for insertion. Dialkylcarbamoylchloride dressing was placed below the subcutaneous anchoring at the time of CVC placement and at each dressing change. Data about the catheters and the exit site were recorded and then compared with an historical cohort. Results: 118 catheters were placed during the studied period. The dialkylcarbamoylchloride dressing was well-tolerated. No case of systemic or local infection was recorded. The comparison with the historical cohort showed a reduction in the rate of exit site infection (*p* value 0.03). Conclusion: Dialkylcarbamoylchloride dressing is well-tolerated in paediatric and neonatal population. It represents a promising tool as a strategy for infection prevention.

## 1. Introduction

Venous access is critical in the care of paediatric and neonatal patients. Venous access devices (VADs) can be divided into central devices (when the tip of the catheter is in the superior vena cava, right atrium or inferior vena cava) or peripheral devices (tip of the catheter in any other location). Central venous catheters (CVCs) should be distinguished according to the recent terminology suggested by the WoCoVA Foundation into PICCs = Peripherally Inserted Central Catheters, or CVCs inserted in the deep veins of the arm (axillary, brachial, basilica); CICCs = Centrally Inserted Central Catheters, or CVCs inserted in the supraclavicular veins (internal jugular, external jugular, subclavian, brachio-cephalic) or infra-clavicular (axillary, cephalic); and FICCs = Femoral Inserted Central Catheters (common femoral, superficial femoral, saphenous). PICCs, CICCs and FICCs are indicated for the infusion of solutions incompatible with the peripheral route, for repeated and frequent blood sampling and for hemodynamic monitoring [1,2].

CVCs should be placed without any exception with the use of US guidance as recommended by current guidelines [3]. Several approaches have been described in this regard [1,4,5,6].

In the choice between PICC, CICC and FICC it is very important to consider the diameter of the vein throughout a preoperative ultrasound scan. The current international guidelines—such as the WoCoVA-GAVeCeLT-WINFOCUS consensus [1]—suggest to measure the diameter of the vein before the insertion of the catheter in order to match the vein diameter with the catheter diameter and reduce the risk of venous thrombosis. It is commonly recommended that the external diameter of the catheter should not exceed 1/3 of the internal diameter of the vein [7]. When the deep veins of the arm are available at the ultrasound examination, the preferable central VAD is the PICC, in children as in adults [8,9]. However, this device cannot be placed in newborns, considering the relatively small veins in the arms of this population [10]. Furthermore, PICCs offer two other significant advantages: (1) PICCs can be placed even in very fragile and instable patients; (2) the exit site of this VAD is favourable for the low bacterial contamination of the skin and the stability itself.

The main contraindications to the placement of a PICC are the presence of a chronic renal failure of grade 3b−4–5 or the bilateral presence of pathologies affecting the upper limbs. In newborns or in paediatric patients, when placement of a PICC is not feasible for the size of the available veins, CICC represents the best choice. However, independently from the type of VAD placed, the care of the exit site after the insertion procedure is the most important thing to be considered in order to minimize complications such as infection, catheter dislodgment and venous thrombosis.

The main purposes in the care of the exit site are: (1) the prevention of extraluminal colonisation, which might lead to a systemic infection (Catheter-Related Blood Stream Infection (CRBSI)); (2) the maintenance of the skin integrity, thus preventing medical adhesive risk injury (MARSI) [3]; (3) the stabilisation of the vascular access device. These issues are particularly important in the neonatal and paediatric population for several reasons, including the fragility of the skin itself, the high rate of dislodgement in this age group and the very high rate of transepidermal water loss, which is very important especially in preterm neonates [11]. Exit-site management includes periodic dressing changes that are carried out every 7–10 days or when dressing appears damp, loose or visibly dirty. This is particularly important as a strategy for infection prevention.

Infections may occur as infections at the exit site or infections of the tunnel or invasive infections (CRBSI). The latter is a severe and possibly life-threatening complication. The treatment of CRBSI depends on several factors, including the microorganism, the type of the vascular device itself, the availability of other vascular access devices for antibiotic administration and so on; however, the removal of the catheter in such cases is a quite often necessary [12,13,14].

Dialkylcarbamoylchloride (DACC) is a hydrophobic fatty acid that can be used to coat the material of a dressing, thus giving hydrophobic properties to the dressing itself. Microorganisms are, therefore, irreversibly bound to the surface of the dressing using hydrophobic interactions [15,16]. The principle of hydrophobic interaction is a key mechanism for bacterial attachment [17,18]. Both in vitro and in vivo tests demonstrate the efficacy of the DACC coating in reducing the bacterial load of the wound and facilitating its healing [19]. Although in vitro results are not necessarily representative of the clinical situation, this property is particularly interesting [20,21], since it can be transferred to the dressing itself. In fact, theoretically, bacteria and fungi become inert once attached to the dressing, avoiding their proliferation and the release of exotoxins and endotoxins [16,22,23].

Our hypothesis is that the use of DACC dressings for the management of the exit site allows for reduction of the bacterial load at each dressing change, since the microorganisms can be removed from the exit site along with the dressing itself. No antiseptic or antibiotics are involved in this process, therefore there is no theoretical risk of bacterial resistance [24].

The aim of this study is to evaluate the feasibility, the safety and the skin tolerance of DACC dressings in the management of the exit site of tunnelled and non-tunnelled CVC in a paediatric and neonatal population. The secondary aim is to estimate the risk of infection compared with an historical cohort.

## 2. Materials and Methods

The study was conducted at the Infermi Hospital in Rimini from September 2020 to December 2022. In our hospital, there is a Neonatal/Paediatric Vascular Access Team (N-*p* TAV) that is responsible for the placement and management of CVCs in the paediatric and neonatal population. All CVCs are inserted according to the SIC Ped protocol [25,26], which includes:Preprocedural ultrasound evaluation (RaCeVA [27], RaPeVA [28] and RaFeVA [29])Appropriate aseptic technique (hand hygiene, maximum barrier protection, skin antisepsis with 2% chlorexidine) [3,12,30]Real time ultrasound guided venepuncture [1]Intra-procedural verification of central position of the tip by non-invasive methods (IC-ECG and RT-US) [31,32,33]Tunnelling according to RAVESTO (Rapid Assessment of Venous Exit Site and Tunnelling Options) [34]Sutureless securement of the catheter or subcutaneous anchoring system [35,36,37]Protection of the exit site with glue [38,39] and semipermeable transparent membrane [40]

Catheters included in the present study are only those secured with a subcutaneous anchoring system (SAS) (SecurAcath™, Interrad Medical, Inc. Plymouth, MN). Fully implantable catheters and central venous catheters stabilized only with tissue adhesive and other sutureless securement were excluded from the study. No antibiotic prophylaxis was used at the time of CVC placement.

In our population, a DACC (Essity, Stockholm, Sweden) dressing was placed around the exit site and below the subcutaneous anchoring system (SAS). The DACC dressing (sterile packaged) includes a four-layer sheet in pre-cut DACC of 4 × 2 cm to be placed in direct contact with the patient’s skin and a small non-woven gauze of the same size, also pre-cut to be placed immediately above the DACC dressing and under the subcutaneous anchoring system (See Figure 1 and Figure 2).

The dressing change was then performed weekly according to international guidelines [2,31]. In particular, exit-site management included:removal of the old dressing (transparent semipermeable membrane) and the DACC dressingcutaneous antisepsis around the exit site with chlorhexidine 2% gluconate in 70% isopropyl alcohol (PAH), using single-dose, disposable and sterile applicators (ChloraPrep™ Becton, Dickinson and Company, Franklin Lakes, NJ, USA)application of a new DACC dressing and a transparent semipermeable membrane with high moist-vapour transmission rate [41]

The aforementioned four-layer DACC leaflet is green in colour and allows a constant indirect inspection of the exit site. In fact, if there are signs of initial infection of the exit site, the dressing changes colour, turning from green to dark brown and then to black. The non-woven gauze is mainly used as an anti-decubitus of the SAS, protecting the fragile skin of paediatric patients from the continuous and constant pressure of the SAS.

Data recorded during the study period included the number of CVC placed, the number of catheter days, exit-site infections reported as a number per 1000 catheter days, CRBSIs reported as a number per 1000 catheter days, local skin injuries, number of dressing changes, reason for catheter removal, main catheter-related complications (such as thrombosis, etc.) and baseline characteristics of the population (age, weight, etc.). All data were collected in the electronic database and then compared with a historical cohort of CVCs inserted from September 2018 to September 2020, before DACC dressing was introduced in our daily practice.

Statistical analysis was performed using MedCalc Statistical Software (version 20.218 MedCalc Software bv, Ostend, Belgium; https://www.medcalc.org; accessed on 26 March 2023). Continuous variables were analysed using Student’s *t* test or Mann–Whitney U test according to their distribution. Categorical variables were analysed using Fisher exact test. A *p* value < 0.05 was considered statistically significant.

Informed consent was obtained from both parents at the time of CVC insertion.

Infections related to CVC were treated according to IDSA guidelines [14].

## 3. Results

From September 2020 to December 2022, 88 CVCs were placed in the paediatric population (46 centrally inserted central catheters (CICCs), 38 peripherally inserted central catheters (PICCs), 4 femoral inserted central catheters (FICCs). 30 CVCs were placed in the neonatal population (23 CICCs and 7 FICCs). Indications for CVC placement were oncologic condition (82%), parenteral nutrition (12%), prolonged antibiotic treatment (4%) and others (2%). Of the 118 CVCs placed, 92 catheters were tunnelled according to the SIC protocol. Mean age of the studied population was 6.8 years. Mean dwell time of the CVCs was 100 days, for a total of 11,151 catheter-days. 1721 dressing changes were performed during the studied period according to our protocol (see Table 1). DACC dressing was well-tolerated in all patients. We had no cases of local infection of the exit site and no cases of CRBSI (0/1000 catheter-days). The exit site always remained clean without any signs of local hyperaemia or infection. We had one case of MARSI in a tunnelled PICC, probably due to the use of a non-breathable cover; in this case, we treated the skin lesion with a DACC dressing, with complete healing after four weeks. We recorded two cases of catheter-related thrombosis (0.18/1000 catheter days), see Table 1.

In the historical cohort, 86 devices were placed and analysed. 39 CICCs, 25 PICCs and 3 FICCs were placed in the paediatric population, and 13 CICCs and 6 FICCs were placed in neonatal patients (see Table 2). Of the 86 devices placed, 73 catheters were tunnelled for a better location of the exit site. Mean age of the studied population was 7.1 years. Mean dwell time was 92 days, for a total of 8195 catheter-days. We recorded four cases of exit-site infection (0.48/1000 catheters days) and two cases of CRBSI (0024/1000 catheter days).

## 4. Discussion

CRBSI is the one of the most important complications related to VADs, since it is associated with high morbidity, loss of the device itself, failure of therapy administration, interruption in the care of patients and increased healthcare costs [12]. The diagnosis of CRBSI is based on the simultaneous blood culture from the catheter and from a peripheral vein. A final diagnosis of CRBSI can be given if the culture of catheter blood becomes positive at least 2 h before the culture of peripheral blood, called the Differential Time to Positivity (DTP) method [14], which is easier and less expensive. Unfortunately, a successful treatment of CRBSI quite often includes the removal of the device itself. It is therefore clear that the prevention of CRBSI has a critical role in the care of paediatric and neonatal patients. The strategy for the prevention of CRBSI is based on the prevention of bacterial contamination by the extraluminal and intraluminal routes. The strategies for preventing extraluminal contamination include:A solid insertion bundle such as the one used in the present study, which includes, among others, the proper choice of the exit site, maximal barrier precautions, the use of 2% chlorexidine in 70% isoprohyl alcohol, the use of SAS, hand hygiene and cyanoacrylate glue at the time of CVC placementA proper policy of dressing change

In addition, the best available evidences to minimize intraluminal contamination have also been applied in the present study, including [42]:Use of needle-free connectors and port protectors, also known as “disinfecting caps”Hand hygieneProper protocol for changing of the administration setsProphylactic lock with taurolidine was also considered in children at high risk for CRBSI, especially if their venous patrimony is scarce and if the intravenous treatment is expected to last for a prolonged time (infants on long parenteral nutrition for short bowel syndrome)

However, not all catheter-related infections are necessarily systemic infections. Sometimes, local infection may occur, such as exit-site infection or tunnel infection, which are usually the results of an uncontrolled bacterial contamination of the skin around the VAD. These conditions might become relevant, since an exit-site infection of a non-tunnelled VAD could quite easily become a CRBSI. On the other hand, when a tunnel infection occurs the removal of the device is inevitable. Exit-site infection diagnosis can be performed by inspection and palpation: in fact, the presence of redness and tenderness of the exit site is highly indicative of a local infection [3]. The treatment of an exit-site infection sometimes requires the removal of the device; however, if the catheter is tunnelled, it is possible to save the VAD using antibiotics (preferably after culturing of the local secretions) and daily antisepsis of the exit site.

It is, therefore, clear that the prevention of exit-site infections is important to prevent CRBSI and is mainly based on strategies aiding the minimization of extraluminal contamination:Proper choice of the exit site. The exit site of a central VAD should be far from areas with high bacterial contamination, such as mouth, urethra, tracheostomy, ileostomy and so on. In addition, the stability of the area should be considered in the choice of the exit site, since an unstable area might be the cause of micro-movements of the device, which increases the risk of extraluminal contamination [34]. The best areas to place the exit site are probably the mid-portion of the upper arm [34], the infraclavicular area and the mid-thigh. Tunnelling is the best technique which allows health care providers to choose the best vein for the venepuncture, and at the same time is the best exit site [28,34,43]. Tunnelling is a powerful technical tool for obtaining the best exit site independently from the venepuncture site. A good example of its usefulness is the insertion of ultrasound guided CICCs in neonates, where the brachio-cephalic vein (BCV) represents the safest and easiest access to the central veins, as has been proven by several papers [44,45,46,47,48,49,50,51]; however, when the BVC is used for the cannulation, it is mandatory to tunnel the catheter in order to move the exit site into the infraclavicular area, which is more stable and has cleaner skin [26]. This technique can be used in virtually any central CVCs as long as a modified Seldinger technique is used for the placement of the device.Proper hand hygiene. Every manipulation of the VADs should be preceded by proper hand hygiene.Skin antisepsis. During VAD insertion and at each dressing change, the skin should be disinfected using 2% chlorhexidine in 70% isopropyl alcohol with a single sterile applicator [3,42,43].Maximal barrier precautions (including cap, mask, sterile gloves, sterile gown and sterile field) as recommended by all guidelines including the recent SHEA published in 2022 [13].Cyanoacrylate glue. The use of Cyanoacrylate glue during CVC insertion is nowadays part of several insertion bundles [25,39]. The glue has three important roles, since it secures the catheter, stops bleeding and has antimicrobial activities. Furthermore, in vitro studies have demonstrated that the long-term use of glue does not alter the chemical and physical properties of polyurethane catheters [51].Securement using sutureless devices. In this regard, SAS must be applied at the time of insertion, and can stay in place as long as the catheter is needed without any routine replacements. Even though SASs are more expensive than other sutureless devices, they are cost-effective, especially when the risk of accidental dislodgment is estimated to be high; this is particularly true in the paediatric population. Recent clinical studies have demonstrated the high effectiveness of SASs in preventing dislodgment in neonates and children, as well as their safety since SASs are well-tolerated even with the fragile skin of preterm neonates [26,36,37].A good, transparent, semipermeable membrane. The exit site must be covered and protected with a transparent semipermeable membrane [3,12,42] which constitutes a real mechanical barrier against environmental bacteria. Sterile semipermeable transparent dressings in polyurethane are the best protection of the exit site and of the area surrounding the emergence of the catheter from the skin, especially when compared to gauze dressing [3,42].Weekly dressing change. Lastly, the exit site is protected throughout a weekly change of the exit site dressing.

It is clear that the effective management of the bacterial load at the exit site is particularly important as a strategy for infection prevention. Good care of the exit site might have an impact on the dwell time of the catheter itself and on patients’ comfort. The DACC technology is a complete novelty in this field, since it can bind bacteria rather than kill it in situ, thus preventing the release of potentially harmful substances (such as endotixin) that might damage the surrounding environment [24,52,53,54]. Patient sensitization, the development of multidrug-resistant pathogens, systemic toxicity and the promotion of an extensive inflammatory response are all very real problems for the clinician in managing a catheter exit-site infection. The DACC dressing is the first dressing to utilize the hydrophobic properties inherent to a wide variety of pathogens, including multi-resistant organisms and biofilms [30], to ensure they are controlled. By facilitating the irreversible binding of microbes to its lining, the DACC dressing is able to provide a safe and effective method of reducing the bacterial load around the device’s emergence site at each dressing change. By guaranteeing control and containment of the bacterial load, the DACC technology can be, in addition to a preventive treatment in the development of an infection, a diagnostic tool since it changes colour as soon as the infection starts, as well as a therapeutic treatment in cases of hyperemia and/or localized exudate at the exit-site level. Being able to balance the bacterial load of the wound in favour of the defence of the organism itself, without risk of cytotoxic reactions or development of bacterial resistance, the DACC dressing should be considered a valid alternative method in the management of the exit site and in the possible treatment of an initial local infection of the exit site, particularly in tunnelled catheters.

Our data suggest that the placement of a DACC dressing below the SAS is completely safe, even in the neonatal population. In the studied population, there was no case of local or systemic infection and no case of skin intolerance. The comparison with the historical cohort proves that the use of DACC dressings might have an impact as a strategy for infection prevention. In fact, in our population, there was a statistically significant reduction in the incidence of exit-site infections (*p* value 0.03).

The strengths of this work are: (1) the large population, (2) the solid insertion bundle that was applied through the whole study (including the historical cohort) and (3) the low rate of infection.

The main weaknesses of the present work are: (1) this is not a randomized controlled trial (comparison with an historical cohort) and (2) catheters stabilized without SAS were excluded.

However, the preliminary results of the present paper might represent the basis for a larger randomized study aimed at evaluating the safety and efficacy of the dialkylcarbamoylchloride dressing compared to current, already standardized exit-site dressing techniques, such as cyanoacrylate glue [39,51] and/or chlorhexidine felt pads [55].

## 5. Conclusions

Our experience comes from the need to manage the exit site effectively, simply and cleanly. In fact, the DACC dressing comes pre-cut in sterile packaging in order to facilitate easy operation as much as possible. The DACC dressing might offer the following advantages:Totally safe in all patients (including newborns)Easy to be handled during dressing changesValid diagnostic tool for any change at the exit site levelBroad compatibility with other exit site management products

DACC dressings are easy to use and well-tolerated by paediatric and/or fragile skin patients and neonates. Their routine use could potentially have an advantage. both in terms of preventing CVC-related infections and in terms of reducing the healing time of any infections localized at the exit site. They represent a promising tool for infection prevention.

## Figures and Tables

**Figure 1 diagnostics-13-01520-f001:**
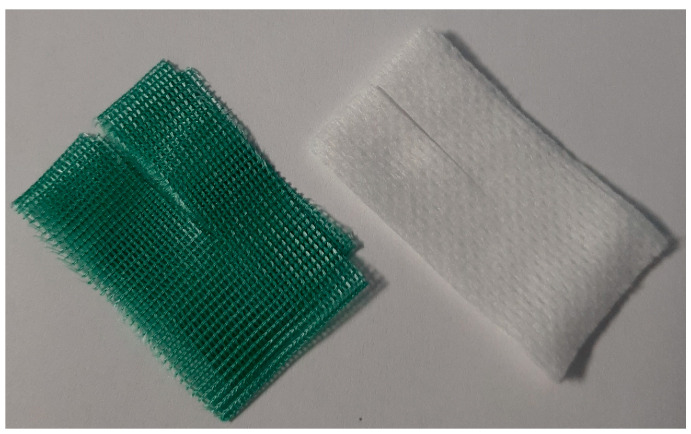
DACC dressing.

**Figure 2 diagnostics-13-01520-f002:**
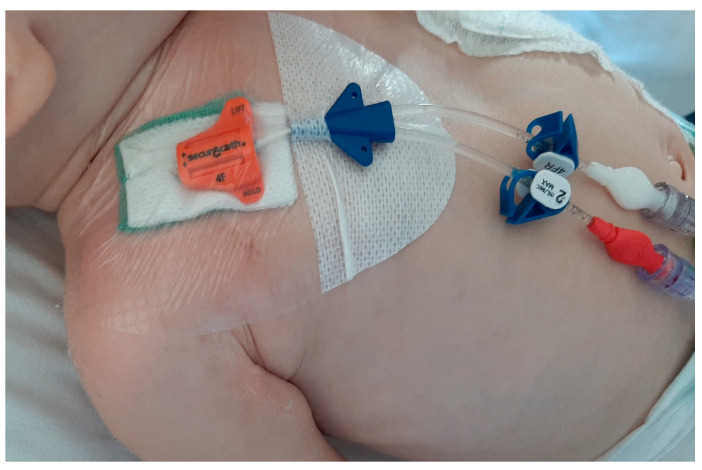
Centrally inserted central catheter in a paediatric patient stabilized with Securacath. DACC dressing is placed below the Securacath.

**Table 1 diagnostics-13-01520-t001:** The table reports the data of the studied population.

	Pediatric Patients	Newborn Patients	Total Patients
Number of catheters	88	30	118
Mean age	7.25 years	38 weeks gestational age	
Mean weight	15.6 kg	2600 kg	
Age min—max	0–17 years	35.1–40.2 weeks gestational age	
M:F	52:36	16:14	68:50
CICCs	46	23	69
PICCs	38	0	38
FICCs	4	7	11
Tunnelled	64	28	92
Mean dwell time (days)	113	28	100
Number of dressings	1578	143	1721
Exit-site infections	0	0	0
CVC-related infections	0	0	0
Thrombosis	1 (1.1%)	0	1 (0.84%)
Other complications	3 (3.4%)	1 (3.5%)	4 (3.3%)
Off-therapy removals	84 (95%)	29 (97%)	113 (97%)

**Table 2 diagnostics-13-01520-t002:** Comparison of the studied population with an historical cohort.

	Studied Population	Historical Cohort	*p* Value
Number of catheters	118	86	-
Mean age	6.8 years ± 2.7	7.1 years ± 2.1	0.3987
Age min–max	0–17	0–17	-
M	68 (57%)	49 (57%)	1.000
CICCs	69 (58%)	52 (60%)	0.8812
PICCs	38 (32%)	25 (29%)	0.6489
FICCs	11 (9.3)	9 (10%)	0.8149
Tunnelled	92 (78%)	73 (85%)	0.2795
Average stay time (days)	100 ± 38	92 ± 29	0.1035
Number of dressings	1721	1370	-
Exit-site infectionss	0	4 (4.6%)	0.0302
CVC-related infection	0	2 (2.3%)	0.1765
Thrombosis	1 (0.8%)	1 (1.2%)	1.000
Other complications	3 (2.5%)	2 (2.4%)	1.000
Off-therapy removals	113 (96%)	77 (89%)	0.0974

## Data Availability

The data presented in this study are available on request from the corresponding author.

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
