# Peer review of "Use of the Dialkylcarbamoylchloride Dressing in the Care of Central Venous Access Exit Site in a Pediatric and Neonatal Population"

_diagnostics, 2023, doi:10.3390/diagnostics13091520_

Round 1

Reviewer 1 Report

Dear authors,

Thank you for presenting your results of efforts to improve the CVC care. 

Dear authors

Thank you for a valuable study. It is well written and of clinical interst. However, there are specific points that need your answering and corrections. This is needed before publication can be accepted.

Method

Could authors please describe the anchoring method used? Which type of stuches were used?

Was any antiobiotic used prophylactic? Which types then? Was the same or different antibiotic approach used in the historical versus the methodological group?

Please add information about the approach step by step for treating an observed infection in the CVC. Also including any possible antibiotic treatment. This is because this could imply the frequency of need of removal, and it could differ between centers.

Results

Please describe the weight of the children receiving the CVC

Please add information about indications for the CVC. E.g. how many patients had it because of hematological disease, oncology, or malnutrition? The frequency for patients receiving it only because of difficulties in peripheral access?

Discussion

The discussion is a bit too long and needs improvement e.g. by including also reflections around the used methodology.

The suggesting points 1-8 could be replaced by a discussion table/figure and referred to.

There are more methods described in the literature to prevent CVC infections. Please discuss pros and cons regarding them and your own used method.

Please discuss limitations of comparing one historical group with a methodological group.

Conclusion

The conclusion is too long and should only include what your results suggest. Therefore, please omit conclusions e.g. regarding bacteria load, acceleration of wound healing, peri inflammatory signs and others that were not studied. Only conclusion based on your own results are of interest.

Author Response

Reviewer 1

Thank you for a valuable study. It is well written and of clinical interest. However, there are specific points that need your answering and corrections. This is needed before publication can be accepted.

Method

Could authors please describe the anchoring method used? Which type of stuches were used?

Thank you for your comment. Has pointed out in line 115 “Catheters included in the present study are only the ones secured with a subcutaneous anchoring system (SAS).” Therefore no stiches was used

Was any antiobiotic used prophylactic? Which types then? Was the same or different antibiotic approach used in the historical versus the methodological group?

Thank you for your comment. This information was added “No antibiotic prophylaxis was used at the time of CVC placement.”

Please add information about the approach step by step for treating an observed infection in the CVC. Also including any possible antibiotic treatment. This is because this could imply the frequency of need of removal, and it could differ between centers.

Thank you for your comment. No case of CVC related infection (CRBSI) was diagnosed in the studied population. However CRBSI would have been treated according to IDSA guideline, this information was added in the paper

Results

Please describe the weight of the children receiving the CVC.

Thank you for your comment. This information was added in table 1

Please add information about indications for the CVC. E.g. how many patients had it because of hematological disease, oncology, or malnutrition? The frequency for patients receiving it only because of difficulties in peripheral access?

Thank you for your comment. This information was added line 161-162

Discussion

The discussion is a bit too long and needs improvement e.g. by including also reflections around the used methodology.

The suggesting points 1-8 could be replaced by a discussion table/figure and referred to.

Thank you for your comment. However, the journal policy suggests to have a well and deep discussion. Therefore, at the moment we decided to leave the discussion as it is unless the editor suggests otherwise.

There are more methods described in the literature to prevent CVC infections. Please discuss pros and cons regarding them and your own used method.

Thank you for your comment all the methods with a strong evidence for the prevention of CVC related infection are discussed in the points 1 to 8 of the discussion. We applied them all. We tested if the DACC could add an extra benefit to what has been already established as effective

Please discuss limitations of comparing one historical group with a methodological group.

Thank you for your comment. This has been added at the line 299

Conclusion

The conclusion is too long and should only include what your results suggest. Therefore, please omit conclusions e.g. regarding bacteria load, acceleration of wound healing, peri inflammatory signs and others that were not studied. Only conclusion based on your own results are of interest.

Thank you for your comment. The conclusion were changed accordingly.

Reviewer 2 Report

I read with great interest the paper by Lamberti et al. on the use of the dialkylcarbamoylchloride dressing in the care of central venous access exit site in a pediatric and neonatal population. The authors found a reduction in the rate of exit site infection with the use of dialkylcarbamoylchloride dressing, which was also well tollerated.

The manuscript is sound and well written. However, I have some minor issues to be addressed.

Abstract

- Please replace "Results.118 catheters" with "Results: 118 catheters"

Introduction

- Beside the use of ultrasound to find the best site for cannulation, you should underline the role of ultrasound guidance for CVC placement in the internal jugular vein (doi: 10.1378/chest.06-2711), subclavian vein (doi: 10.1097/CCM.0000000000005819) and brachiocephalic vein (doi: 10.1016/j.redare.2020.10.011). Please briefly discuss and add these 3 references.

Methods

- When you report the SIC Ped protocol, please specify whether the ultrasound guidance was a real-time guidance.

- Please report the Ethical Committee approval in the methods section.

Author Response

Reviewer 2

I read with great interest the paper by Lamberti et al. on the use of the dialkylcarbamoylchloride dressing in the care of central venous access exit site in a pediatric and neonatal population. The authors found a reduction in the rate of exit site infection with the use of dialkylcarbamoylchloride dressing, which was also well tollerated.

The manuscript is sound and well written. However, I have some minor issues to be addressed.

Abstract

- Please replace "Results.118 catheters" with "Results: 118 catheters"

Thank you for your comment. The manuscript was changed accordingly

Introduction

- Beside the use of ultrasound to find the best site for cannulation, you should underline the role of ultrasound guidance for CVC placement in the internal jugular vein (doi: 10.1378/chest.06-2711), subclavian vein (doi: 10.1097/CCM.0000000000005819) and brachiocephalic vein (doi: 10.1016/j.redare.2020.10.011). Please briefly discuss and add these 3 references.

Thank you for your comment. The manuscript was changed accordingly and the references added

Methods

- When you report the SIC Ped protocol, please specify whether the ultrasound guidance was a real-time guidance.

Thank you for your comment. The manuscript was changed accordingly